# Turbulent Processes in the Earth's Magnetotail: Spectral and Statistical Research

Liudmyla V. Kozak[1,2], Bohdan A. Petrenko[1], Anthony T.Y. Lui[3], Elena A. Kronberg[4,5], Elena E. Grigorenko[6], and Andrew S. Prokhorenkov[1]

[1]Taras Shevchenko National University of Kyiv, Kyiv, Ukraine
[2]Space Research Institute of the National Academy of Sciences of Ukraine and State Space Agency of Ukraine, Kyiv, Ukraine
[3]Johns Hopkins University Applied Physics Laboratory, Laurel MD, USA
[4]Max Planck Institute for Solar System Research, Göttingen, Germany
[5]Department of Earth and Environmental Sciences, Ludwig Maximilian University of Munich, Munich, Germany
[6]Space Research Institute, RAS, Russia

**Correspondence:** PUT

**Abstract.** We use the magnetic field measurements from four spacecraft of Cluster-II mission (3 events from 2005 to 2015) for the analysis of turbulent processes in the Earth's magnetotail. For this study we conduct the spectral, wavelet and statistical analysis. In the framework of statistical examination, we determine the kurtosis for selected events and conduct extended self-similarity evaluation (analysis of distribution function moments of magnetic field fluctuations on different scales). We compare high order structure function of magnetic fluctuations during dipolarization with isotropic Kolmogorov and three-dimensional log-Poisson model with She-Leveque parameters. We obtain power law scaling of the generalized diffusion coefficient (the power index that varies within the range of $0.2 - 0.7$). The obtained results show the presence of super-diffusion processes. We find the significant difference of the spectral indices for the intervals before and during the dipolarization. Before dipolarization the spectral index lies in the range from $-1.68 \pm 0.05$ to $-2.08 \pm 0.05$ ($\sim 5/3$ according to the Kolmogorov model). During dipolarization the type of turbulent motion changes: on large time-scales the turbulent flow is close to the homogeneous models of Kolmogorov and Iroshnikov-Kraichnan (the spectral index lies in the range from $-2.20$ to $-1.53$), and at smaller time scales the spectral index in the range from $-2.89$ to $-2.35$ (the Hall-MHD model). The kink frequency is less or close to the average value of the proton gyrofrequency.

The wavelet analysis shows the presence of both direct and inverse cascade processes, which indicates the possibility of self-organization processes, as well as the presence of Pc pulsations.

## 1  Introduction

The physical process responsible for the onset of magnetospheric substorms remains an unsolved mystery in spite of more than five decades of intense research efforts after the discovery of this episodic disturbance in the ionosphere and the magnetosphere.

Many potential processes have been proposed by e.g. Nishida and Hones (1974); Rostoker and Eastman (1987); Samson (1998); Rothwell et al. (1988); Lui et al. (1991); Haerendel (1992); Kan (1998); Streltsov et al. (2010). Soon after the turn of the century, two prominent scenarios of substorm development emerged with different emphasis on the initial subsotrm onset location and the associated physical mechanism (Baker et al., 1996). The first model is the Near-Earth Neutral Line (NENL) model with the onset location in the middle of the tail at distances of 15-30 Earth radii in which a large-scale process involving reconnection of magnetic field lines is invoked by Baker et al. (1996); Nishida (1978). The second one is the Current Disruption (CD) model in which a plasma instability at distances of 6-15 Earth radii is invoked initially by Lui (1991); Roux et al. (1991); Samson (1998), followed by magnetic reconnection at further downtail distances (Lui, 1991). The distinguishing characteristics of these two scenarios are the initial onset location and the associated physical process.

A four-satellite ESA mission, named Cluster-II and a five-satellite NASA mission, named Time History of Events and Macroscale Interactions during Substorms (THEMIS), were launched to identify the location where the substorm disturbances are initiated in the magnetotail (Angelopoulos, 2008).

The strategy adopted by these missions is to have some satellites situated at different downtail distances to identify the originating location of substorm disturbance. This strategy turns out not to be foolproof as magnetic reconnection was later recognized to be localized in the local time extent and not a large-scale process as originally envisioned (Nakamura et al., 2004). Because of the spatial limitation of magnetic reconnection, satellite observations have not led to a compelling conclusion to settle the mystery as observations on the propagation direction of substorm disturbances in the tail yielded diversified results with many reports on results to be consistent with one or the other of the scenarios, i.e., no consistency with one particular model (Angelopoulos, 2008; Lui, 2009; Akasofu, 2012, 2017; Panov et al., 2013; Hwang et al., 2014). A complication in distinguishing the two scenarios is the presence of the so-called pseudo-breakups (Zelenyi and Veselovskiy, 2008; Lopez, 1990; Lui, 2002, 2004; Runov et al., 2012).

On the other hand, both scenarios have common consequences such as impulsive particle acceleration, dipolarization, and formation of a current wedge (Zelenyi and Veselovskiy, 2008; Lui, 2004). Several plasma instabilities have been proposed to play a role in these substorm scenarios. External and internal plasma environments with the presence of heavy ions affect the occurrence of these instabilities. Instabilities in the CD model includes the ballooning instability (Roux et al., 1991; Cheng and Lui, 1998) and the cross-field current instability (Lui, 2004). Although magnetic reconnection is not a plasma instability process, it requires an instability such as ion tearing instability (Schindler, 1974; Sitnov and Schindler, 2010) to form an X-line for its existence. Besides, magnetic reconnection can involve turbulence, but one should not forget about the work of Speiser (1970) describing themagnetic reconnection without noise (i.e., turbulence). Heavy ions play a significant role in the development of substorms since their presence changes current sheet thickness and its structure, leading to favorable conditions for magnetic reconnection and the generation of Kelvin-Helmholtz instability (Kronberg et al., 2014, 2017a, b).

Investigation of the magnetotail is significantly complicated by the presence of turbulence due to instability resulting in a "catastrophic" alteration of the flow and magnetic field structure (Barenblatt, 2004; Frik, 1999; Frisch, 1995). Complex turbulent processes that occur in the Earth's magnetosphere cannot be described within the analytical MHD flow models. To consider the properties of turbulence at different temporal and spatial scales, one should adopt methods of statistical physics and

the cascade model developed in hydrodynamic theories. Also note that, when considering a statistical system to be characterized by self-similarity, it can be regarded as a physical characteristic of a fractal size equal to the effective Larmor radius of particles and properties of turbulent processes associated not only with the physical mechanisms of instability, but also with symmetries that describe the scale invariance (Savin et al., 2011; Chen et al., 2017).

An analytical or numerical solution of the turbulent plasma dynamics (in 3-dimensional geometry) and determination of turbulence features at large time scales are not currently possible. Therefore, statistical properties of turbulence associated with large-scale invariance are determined experimentally along with estimation of spectral indices in the assumption of power laws for plasma parameters. This allows one to get an idea of the physical properties of plasma turbulence and a description of the transport processes in the turbulent regions in qualitative and quantitative terms (Kozak et al., 2012; Hadid et al., 2015; Kozak

et al., 2015). This approach has yielded important insights on the turbulent plasma characteristics mainly in the magnetosheath. Plasma turbulence in the magnetotail is a key feature for dipolarization in the CD model. The multiscale nature of plasma turbulence at a CD site has also been explored by analysis from the nonlinear dynamics approach or from wave identification by e.g. Lui and Najmi (1997); Consolini and Lui (1999, 2000); Lui (2002); Consolini (2005); Lui et al. (2008); Yoon et al. (2009); Contel et al. (2009); Zhou et al. (2009); Mok et al. (2010).

In this work, the spectral and statistical approach was carried out to examine the features of the magnetic field dipolarization in the Earth's magnetotail for 3 events (2015-09-12, 2005-10-15, 2005-10-01). The methods and approaches used in the work are described in detail and tested in the works by Kozak and Lui (2008); Kozak et al. (2011); Savin et al. (2011); Kozak et al. (2012); Savin et al. (2014); Kozak et al. (2015, 2017); Kronberg et al. (2017a). Acceleration processes of protons and electrons associated with wave activity observed during dipolarization events in 2005 were previously studied in papers by Grigorenko

et al. (2016). This work provides the statistical review estimates on the features of turbulent and dynamic processes at small time scales.

## 2    Used experimental data

The data of the magnetic field for this analysis were obtained by the spacecraft (SC) of the "Cluster-II" mission in the near-Earth tail for 3 events (two events in 2005 and one event in 2015) during the dipolarization of the magnetic field (see Figs. 1a

and 1b). The sampling rate is 22.5 Hz. The magnetic field data are obtained by the fluxgate magnetometer (FGM) (Balogh et al., 2001). In the course of the study, the peculiarities were considered of the magnetic field fluctuations for moments prior to dipolarization (relative level of fluctuations $\sim 0.05 - 0.24$ (interval 1) and during the dipolarization of the magnetic field (relative fluctuation level $\sim 0.8 - 1$ (interval 2) (Figs. 1a and 1b). The spacecrafts were at the geocentric distances 11-17 $R_E$ in anti-sunward direction in the pre-midnight sector (Fig. 2).

The event of 2015 satisfies the most the conditions of dipolarization for the CD model by Lui (2018). For the CD model, large magnetic fluctuations predominantly occur around the neutral sheet of the magnetotail where $B_z \gg B_x, B_y$. During CD, the level of magnetic fluctuations $dB_z/B_{z_0}$ can reach the order of one or more, where $B_{z_0}$ is the $B_z$ value before CD onset. This type of events typically lasts for several minutes. The $B_z$ component could become negative, in spite of a strong

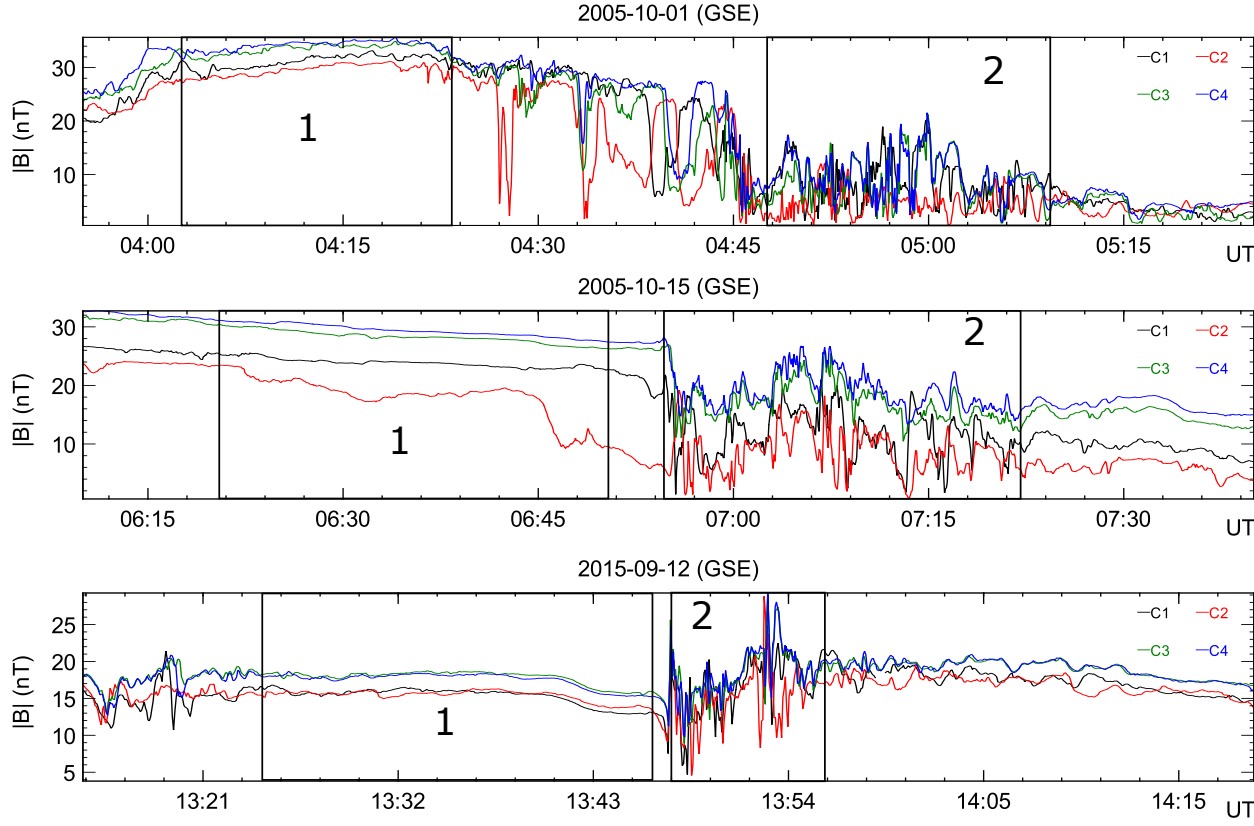

**Figure 1a.** Absolute values of magnetic field in GSE. 1 — intervals for the moments before dipolarization; 2 — intervals during the dipolarization of the magnetic field

background positive $B_z$ component from the dipole magnetic field. It is accompanied by particle energization and intense fluctuating electric fields. The cross-tail current breaks up into filaments and may reverse its direction. The associated plasma flow pattern is not organized by the $B_z$ polarity, unlike magnetic reconnection.

In the dipolarization region the fluctuations of magnetic field greatly differ from the region before dipolarization: in particular for event on 2005-10-01 the magnetic field variations normalized to the current mean value are $\delta B_x/B_x \sim 0.5-1$, $\delta B_y/B_y \sim 1$, $\delta B_z/B_z \sim 1$, $\delta B/B \sim 0.8-1$; for event on 2005-10-15 — $\delta B_x/B_x \sim 0.2-0.5$, $\delta B_y/B_y \sim 0.3-1$, $\delta B_z/B_z \sim 0.4-0.8$, $\delta B/B \sim 0.5-1$; for event on 2015-09-12 — $\delta B_x/B_x \sim 0.5-1$, $\delta B_y/B_y \sim 0.5-0.7$, $\delta B_z/B_z \sim 0.8-1$, $\delta B/B \sim 0.8-1$.

Since the region of dipolarization is traced by four space vehicles, we were able to estimate the speed and direction of the dipolarization front (DF) motion, the thickness of the front (Table 1). The estimated values of plasma characteristics in dipolarization region (interval 2) are collected in Table 2.

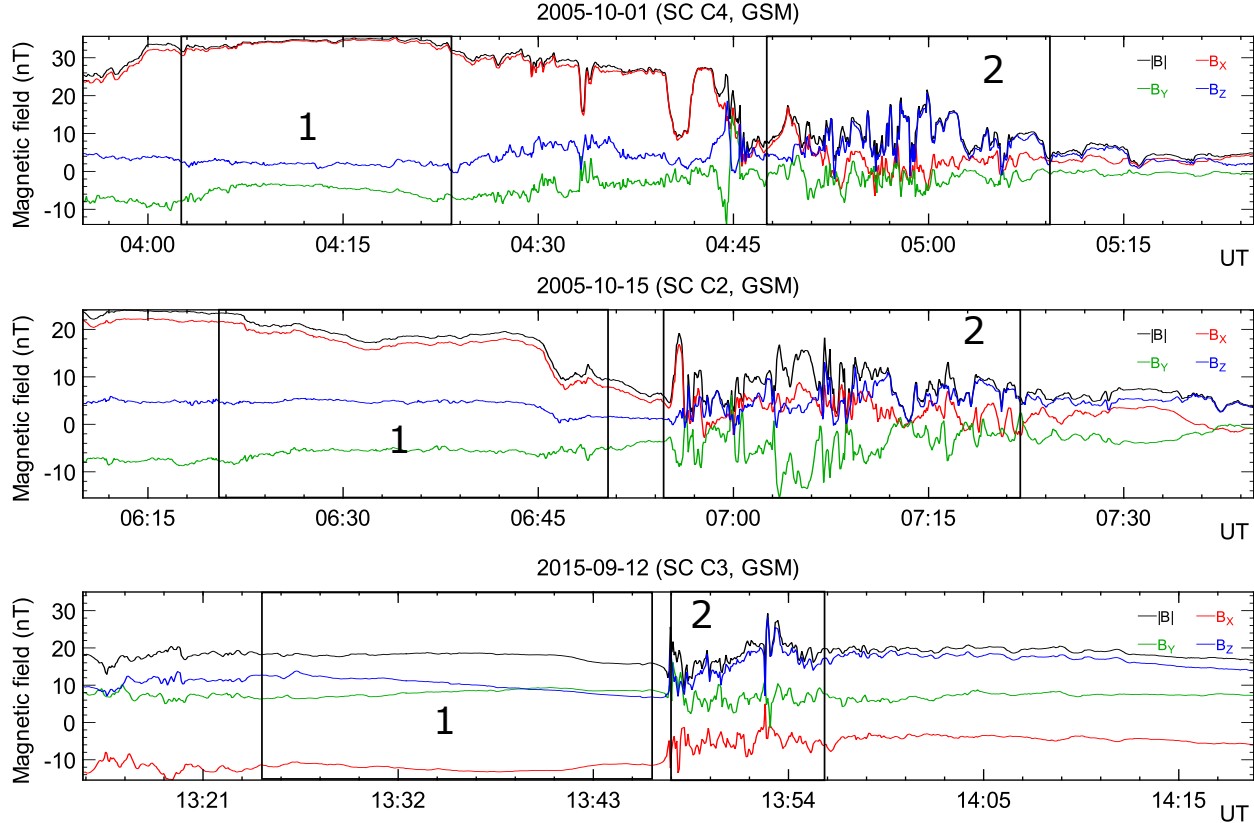

**Figure 1b.** Examples of magnetic field dipolarizations in GSM. The observations are shown from satellites closest to the current layer. 1 — intervals for the moments before dipolarization; 2 — intervals during the dipolarization of the magnetic field

Moreover, according to Fu et al. (2012), during the dipolarization the variation of $B_z$ for different satellites can be represented as:

$$B_{\text{fit}} = \frac{a}{2} \tanh\left(\frac{\Delta t}{b/2}\right) + \left(c + \frac{a}{2}\right) \tag{1}$$

where, $\Delta t = t - t_{DF}$ represents the interval from 60 s before to 15 s after the dipolarization front. $a$, $b$, $c$ are fitting coefficients, and $\sigma$ is a standard error.

The calculated values of the coefficients are also given in Table 1.

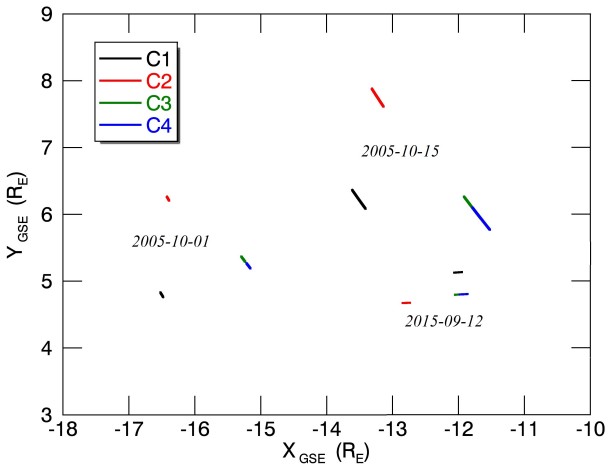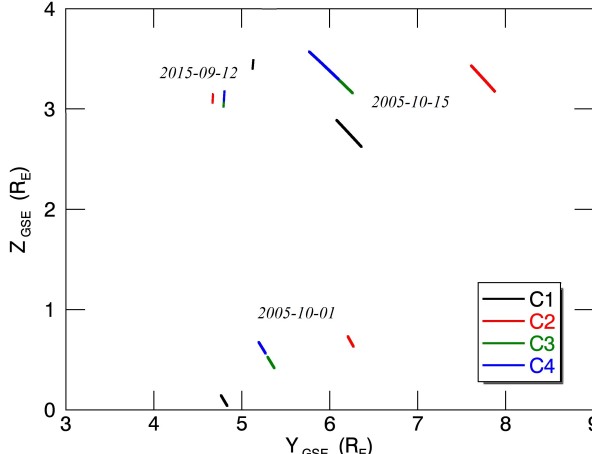

**Figure 2.** The locations of the satellites

**Table 1.** Features of the dipolarization fronts (DFs)

| | | Time of passage of DF | Location | | | Amplitude of the DF, $a$, nT | Front duration, $b$, sec | Standard error of the fitting $\sigma$, nT | The speed of the DF, $V_{\mathrm{DF}}$, km/s | The thickness of the DF, $d$, km |
|---|---|---|---|---|---|---|---|---|---|---|
| | | | $X_{\mathrm{GSE}}$, $R_{\mathrm{E}}$ | $Y_{\mathrm{GSE}}$, $R_{\mathrm{E}}$ | $Z_{\mathrm{GSE}}$, $R_{\mathrm{E}}$ | | | | | |
| Fu et al. (2012) | | | | | | $a > 4$ | $b < 8$ | $\sigma < 2.5$ | 210 (to the Earth) | |
| 2015-09-21 | C1 | 13:47:39 | -12.0675 | 5.1244 | 3.4005 | 8.89 | 0.27 | 1.56 | 350 (to the Earth) | 95 |
| | C2 | 13:47:25 | -12.8409 | 4.6692 | 3.0676 | 9.20 | 0.81 | 1.08 | | 284 |
| | C3 | 13:47:30 | -12.0455 | 4.7940 | 3.0293 | 9.69 | 2.31 | 1.29 | | 809 |
| | C4 | 13:47:31 | -11.9729 | 4.7978 | 3.0873 | 9.80 | 1.77 | 1.42 | | 620 |
| 2005-10-15 | C1 | 06:55:29 | -13.4200 | 6.0945 | 2.8734 | 4.59 | 0.37 | 3.67 | 284 (to the Earth) | 105 |
| | C2 | 06:56:06 | -13.1518 | 7.6310 | 3.4116 | 2.16 | 0.45 | 1.08 | | 128 |
| | C3 | 06:56:00 | -11.6814 | 5.9581 | 3.4140 | 7.93 | 3.72 | 2.13 | | 1056 |
| | C4 | 06:56:00 | -11.5448 | 5.7908 | 3.5521 | 7.54 | 32.62 | 1.03 | | - |
| 2005-10-01 | C1 | 04:44:16 | -16.4133 | 4.6566 | 0.2978 | 4.56 | 0.22 | 2.09 | 208 (to the Earth) | 46 |
| | C2 | 04:43:28 | -16.3271 | 6.1018 | 0.8937 | 4.02 | 1.33 | 3.37 | | 277 |
| | C3 | 04:44:19 | -15.1508 | 5.1772 | 0.6914 | 9.08 | 2.81 | 2.15 | | 584 |
| | C4 | 04:44:23 | -15.0671 | 5.0758 | 0.6914 | 4.53 | 0.71 | 1.38 | | 148 |

**Table 2.** Estimated values of plasma characteristics in dipolarization region

| | SC | Average proton-cyclotron frequency $\langle f_{C_p} \rangle$, Hz | Concentration of electrons, $n_e$, $(cm^{-3})$ | Electron plasma frequency, $\omega_{pe}$, $(s^{-1})$ | Ion plasma frequency, $\omega_{pi}$, $(s^{-1})$ | Electron inertial length, $\lambda_e$, $(km^{-1})$ | Ion inertial length, $\lambda_i$, $(km^{-1})$ |
|---|---|---|---|---|---|---|---|
| 2015-09-12 | C1 | 0.25 | 0.25 | 2.82E+04 | 6.58E+02 | 10.63 | 455.52 |
| | C2 | 0.22 | 0.2 | 2.52E+04 | 5.89E+02 | 11.89 | 509.29 |
| | C3 | 0.28 | 0.35 | 3.34E+04 | 7.79E+02 | 8.98 | 384.99 |
| | C4 | 0.28 | 0.2 | 2.52E+04 | 5.89E+02 | 11.89 | 509.29 |
| 2005-10-15 | C1 | 0.19 | 0.5 | 3.99E+04 | 9.31E+02 | 7.52 | 322.1 |
| | C2 | 0.13 | 0.5 | 3.99E+04 | 9.31E+02 | 7.52 | 322.1 |
| | C3 | 0.27 | 0.5 | 3.99E+04 | 9.31E+02 | 7.52 | 322.1 |
| | C4 | 0.3 | 0.5 | 3.99E+04 | 9.31E+02 | 7.52 | 322.1 |
| 2005-10-01 | C1 | 0.13 | 0.4 | 3.57E+04 | 8.32E+02 | 8.4 | 360.12 |
| | C2 | 0.07 | 0.4 | 3.57E+04 | 8.32E+02 | 8.4 | 360.12 |
| | C3 | 0.14 | 0.4 | 3.57E+04 | 8.32E+02 | 8.4 | 360.12 |
| | C4 | 0.16 | 0.4 | 3.57E+04 | 8.32E+02 | 8.4 | 360.12 |

## 3  Results of the Research

### 3.1  Spectral analysis.

Within the spectral analysis, the spectral power density (PSD) was built from the frequency $f$, and the power-law dependence $\mathrm{PSD}(f) \sim f^a$ was determined. To determine the PSD signal for a series of $N$ measurements $X_n$, a discrete Fourier transform (Daly and Paschmann, 2000) was used:

$$\mathrm{PSD} = \frac{2N}{f_s} \left| \frac{1}{N} \sum_{n=0}^{N-1} X_n \exp\left( \frac{2\pi i n j}{N} \right) \right|^2 \tag{2}$$

where $n = 0, 1 \ldots N-1$, $j = 0, 1 \ldots N/2$.

To find the break points and the slope of the spectrum, we used a piecewise linear approximation of $\log \mathrm{PSD}$ from $\log(f)$ in the frequency range $0.005 - \sim 1.0$ Hz for the pre-dipolarization interval and $0.01 - 3.0$ (events 2005-10-01, 2005-10-15) and $0.01 - 1.0$ (event 2015-09-12) Hz for dipolarization. The limitation of frequencies at high level is due to the presence of instrumental noise, and at low frequency range due to the amount of data sampling and the edge effect of the smoothing procedure. The PSD results for the absolute value of the magnetic field are shown in Fig. 3 and Table 3.

**Table 3.** The results of PSD analysis

| Event | SC | Interval 1 | | Interval 2 | | | | |
|---|---|---|---|---|---|---|---|---|
| | | Slope | Average slope | Kink frequency, f*, Hz | Slope lower f* | Average slope lower f* | Slope higher f* | Average slope lower f* |
| 2015-09-12 | C1 | $-1.7535\pm0.022$ | $-1.86\pm0.10$ | 0.14 | $-1.6340\pm0.031$ | $-1.59\pm0.07$ | $-2.8914\pm0.038$ | $-2.77\pm0.20$ |
| | C2 | $-1.8758\pm0.036$ | | 0.12 | $-1.6619\pm0.034$ | | $-2.4966\pm0.04$ | |
| | C3 | $-1.8722\pm0.036$ | | 0.15 | $-1.5310\pm0.042$ | | $-2.8527\pm0.033$ | |
| | C4 | $-1.9552\pm0.046$ | | 0.15 | $-1.5421\pm0.044$ | | $-2.8473\pm0.029$ | |
| 2005-10-15 | C1 | $-1.6800\pm0.017$ | $-1.86\pm0.16$ | 0.19 | $-2.1634\pm0.020$ | $-2.03\pm0.33$ | $-2.5265\pm0.036$ | $-2.50\pm0.23$ |
| | C2 | $-2.0042\pm0.019$ | | 0.07 | $-1.5395\pm0.026$ | | $-2.7969\pm0.016$ | |
| | C3 | $-1.8452\pm0.023$ | | 0.08 | $-2.1995\pm0.028$ | | $-2.3461\pm0.023$ | |
| | C4 | $-1.9003\pm0.045$ | | 0.08 | $-2.1992\pm0.044$ | | $-2.3504\pm0.047$ | |
| 2005-10-01 | C1 | $-2.0794\pm0.034$ | $-2.04\pm0.04$ | 0.13 | $-1.6442\pm0.026$ | $-1.66\pm0.18$ | $-2.8159\pm0.045$ | $-2.73\pm0.16$ |
| | C2 | $-2.0237\pm0.046$ | | 0.07 | $-1.8831\pm0.031$ | | $-2.4860\pm0.048$ | |
| | C3 | $-1.9987\pm0.026$ | | 0.08 | $-1.5828\pm0.033$ | | $-2.8022\pm0.035$ | |
| | C4 | $-2.0665\pm0.038$ | | 0.1 | $-1.5261\pm0.045$ | | $-2.8221\pm0.032$ | |

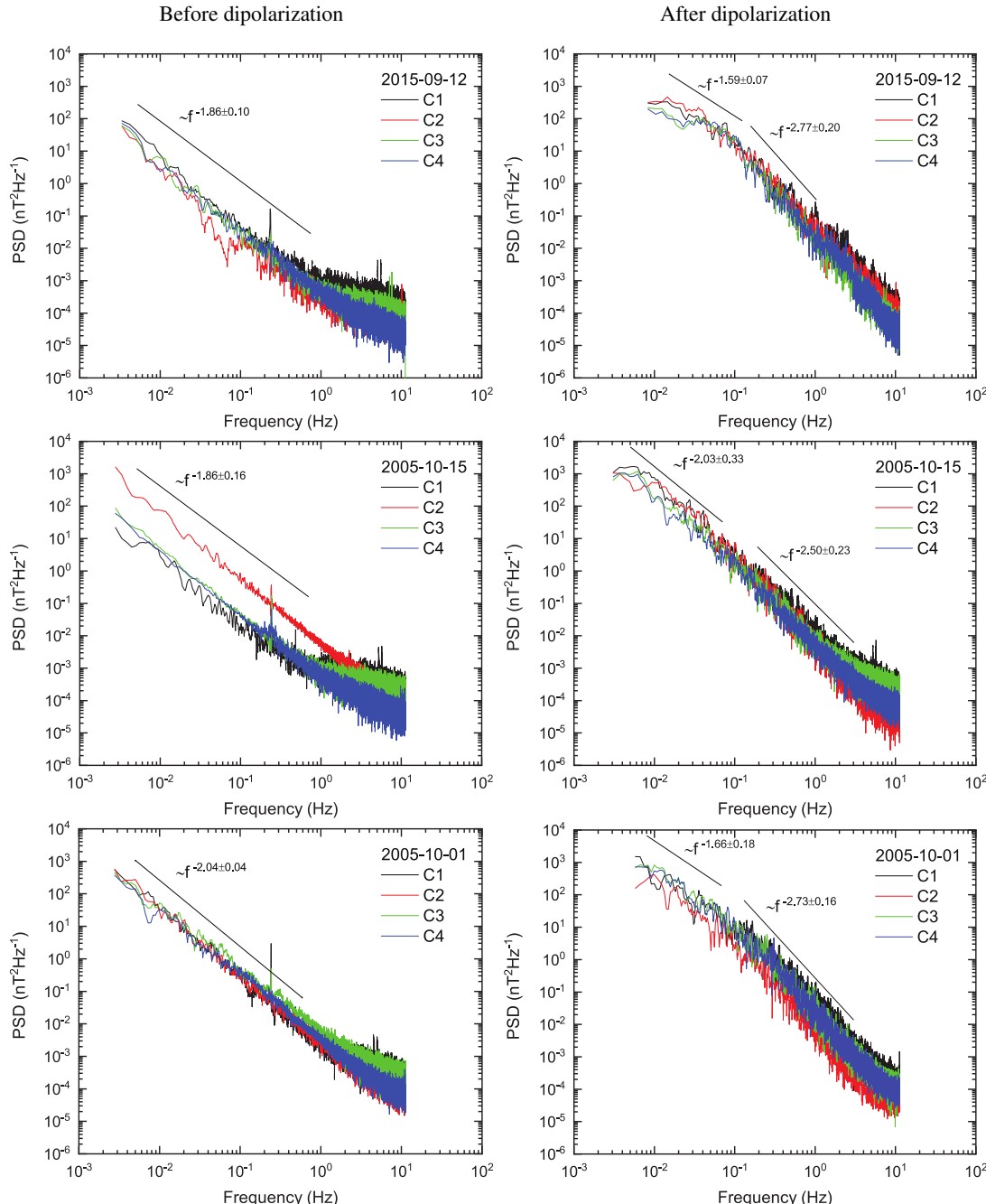

**Figure 3.** The results of spectral analysis

During time before the dipolarization (interval 1), for all events and spacecrafts, there is no sharp change in the PSD power law in the inertial interval (the exponent varies in the range from -2.08 to -1.68). During dipolarization (interval 2), the situation is significantly different. There is an increase in the "steepness" of PSDs for higher frequencies than the kink frequency, which means more efficient energy transfer from large to smaller scales. For practically all spectra of interval 2, the kink frequency is less or close to the average value of the proton gyrofrequency (Table 2). The kink frequency determines the characteristic frequency of the type change (i.e., the energy transfer rate) of the turbulent cascade in the inertial range. In particular, for events 2015-09-12, 2005-10-15 the break corresponds to about half of the proton frequency $-\omega_c/2$. The fact that the break is observed at frequencies smaller than the proton gyrofrequency may indicate a significant effect of heavy ions at the distances considered (according to the measurements of the density by the CIS instrument (Rème et al., 2001) for the event 2005-10-01, in the region of the magnetic field dipolarization, the percentage of oxygen ions in relation to protons ($\langle n(O^+)\rangle/\langle n(H^+)\rangle$) is $21.1\pm10.0\%$ (SC C3) and $9.3\pm1.5\%$ (SC C4), and the percentage of helium in relation to protons ($\langle n(He^+)\rangle/\langle n(H^+)\rangle$) $\sim 2.4\pm0.3\%$ (SC C3) and $\sim 4.8\pm0.7\%$ (SC C4); for the event 2005-10-15 — $\langle n(O^+)\rangle/\langle n(H^+)\rangle \sim 11.1\pm1.0\%$ (SC C4), and $\langle n(He^+)\rangle/\langle n(H^+)\rangle \sim 3.4\pm0.5\%$ (SC C4); for the 2015-09-12 event — $\langle n(O^+)\rangle/\langle n(H^+)\rangle \sim 18.9\pm7.3\%$ (SC C4), and $\langle n(He^+)\rangle/\langle n(H^+)\rangle \sim 15.8\pm5.4\%$ (SC C4)). At the same time, the exponent lies in the range from $-2.2$ to $-1.53$ on large time scales of $0.01-\omega_c/2$, and at smaller time scales $\omega_c/2-3$ Hz, the value lies in the range from $-2.89$ to $-2.35$. The greatest difference at different time scales is observed for the 2015 event.

## 3.2 Wavelet analysis.

Within the framework of the wavelet analysis for a series of measurements $X_n$ ($n=0,1\ldots N-1$) with time shift $\delta t$, a Morlet wavelet (Torrence and Compo, 1998) was used:

$$\Psi_0(\eta) = \pi^{-\frac{1}{4}} e^{i\omega_0\eta} e^{-\frac{\eta^2}{2}} \tag{3}$$

where $\omega_0$ — dimensionless frequency, $\eta$ — dimensionless time.

The continuous wavelet transforms of the discrete signal $X_n$ is defined as the convolution of the mother wavelet whose argument is scaled and transmitted with a signal (Farge, 1992; Grinsted et al., 2004; Jevrejeva et al., 2003):

$$W_n(s) = \sum_{n'}^{N-1} x_{n'} \Psi^* \left[ \frac{(n'-n)\delta t}{s} \right] \tag{4}$$

where (*) — complex conjugate, $|W_n(s)|^2$ — wavelet power spectrum, $s$ — wavelet scale. Index 0 in $\Psi_0$ notes that the function is normalized.

The results of the continuous wavelet transform of the magnetic field module in the dipolarization region are shown in Figs. 4 to 6. The time range was chosen to include the dipolarization interval (interval 2) with some margin ($\pm5$ min) to exclude the influence of the edge effects of the wavelet transform to the explored intervals. The upper limit of the wavelet transform is

limited by the Nyquist frequency. The sampling frequency of the measurements makes it possible to analyze the presence of high-frequency fluctuations in addition to the low-frequency components.

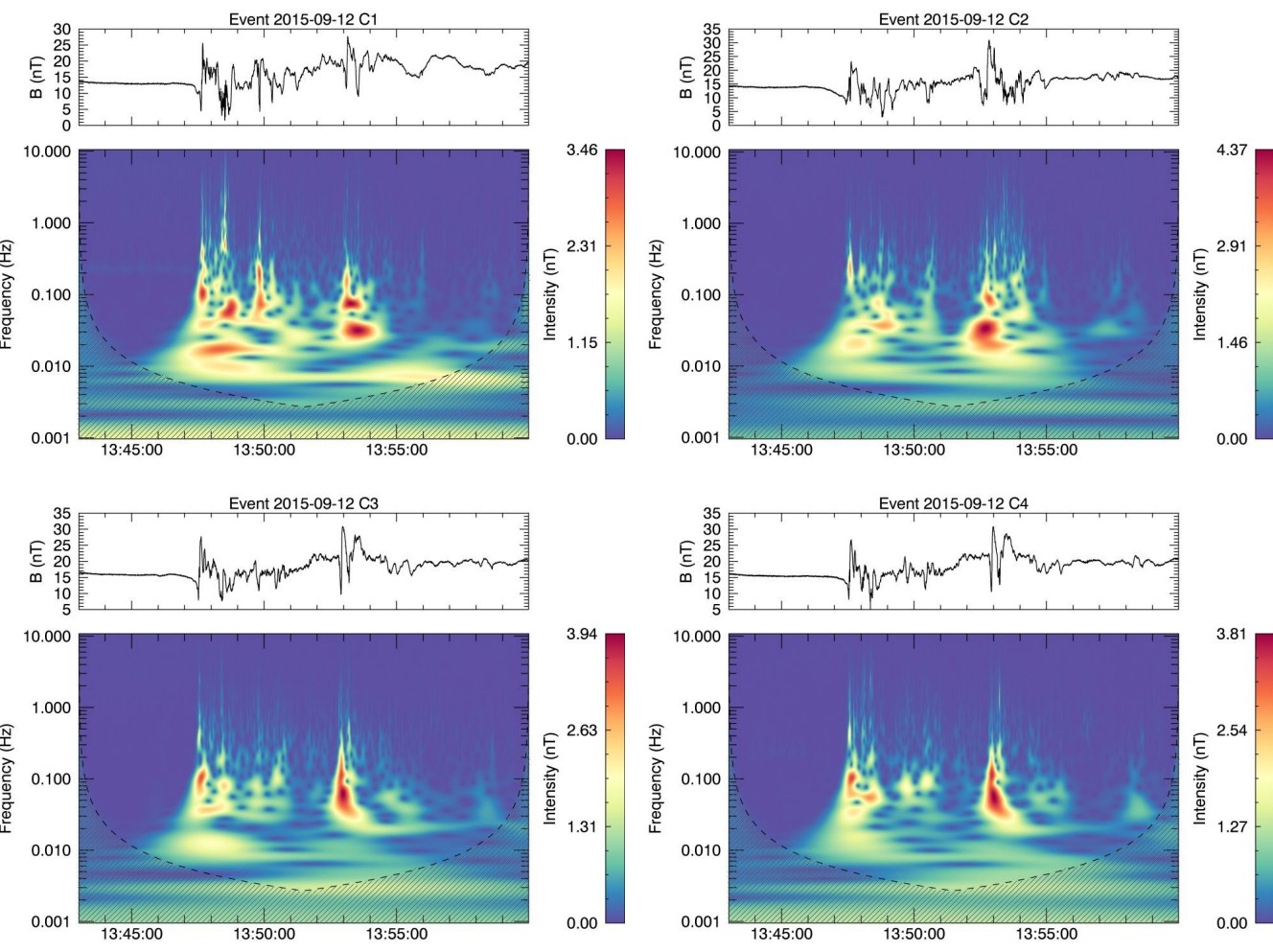

**Figure 4.** The results of wavelet analysis for event 2015-09-12. The cone of influence is shown by shaded region

In Fig. 4 the wavelet analysis of magnetic field magnitude for the event on 2015-09-12 is presented. In this case C3 and C4 were located ahead of the C1 and C2, with C2 being the furthest in the magnetotail. Inverse and direct cascades are present in wavelet analysis at multiple times: 13:47:30 (dipolarization onset) and 13:53:00, both spanning 0.02–0.2 Hz in the frequency domain. This signal broadens for C1 and C2 wavelet and breaks up into smaller time-frequency forms: e.g. the signal on 13:53:00 UT becomes stronger in time and frequency domains of 1 minute and 0.01 Hz correspondingly. Wavelet transform for C1 is characterized by a prevalence of intensity enhancements in wide frequency range at an earlier stage of the turbulent phase of dipolarization at 13:47:30-13:50:30 as compared to transforms for C2, C3, C4. Also for C1, it is interesting to note

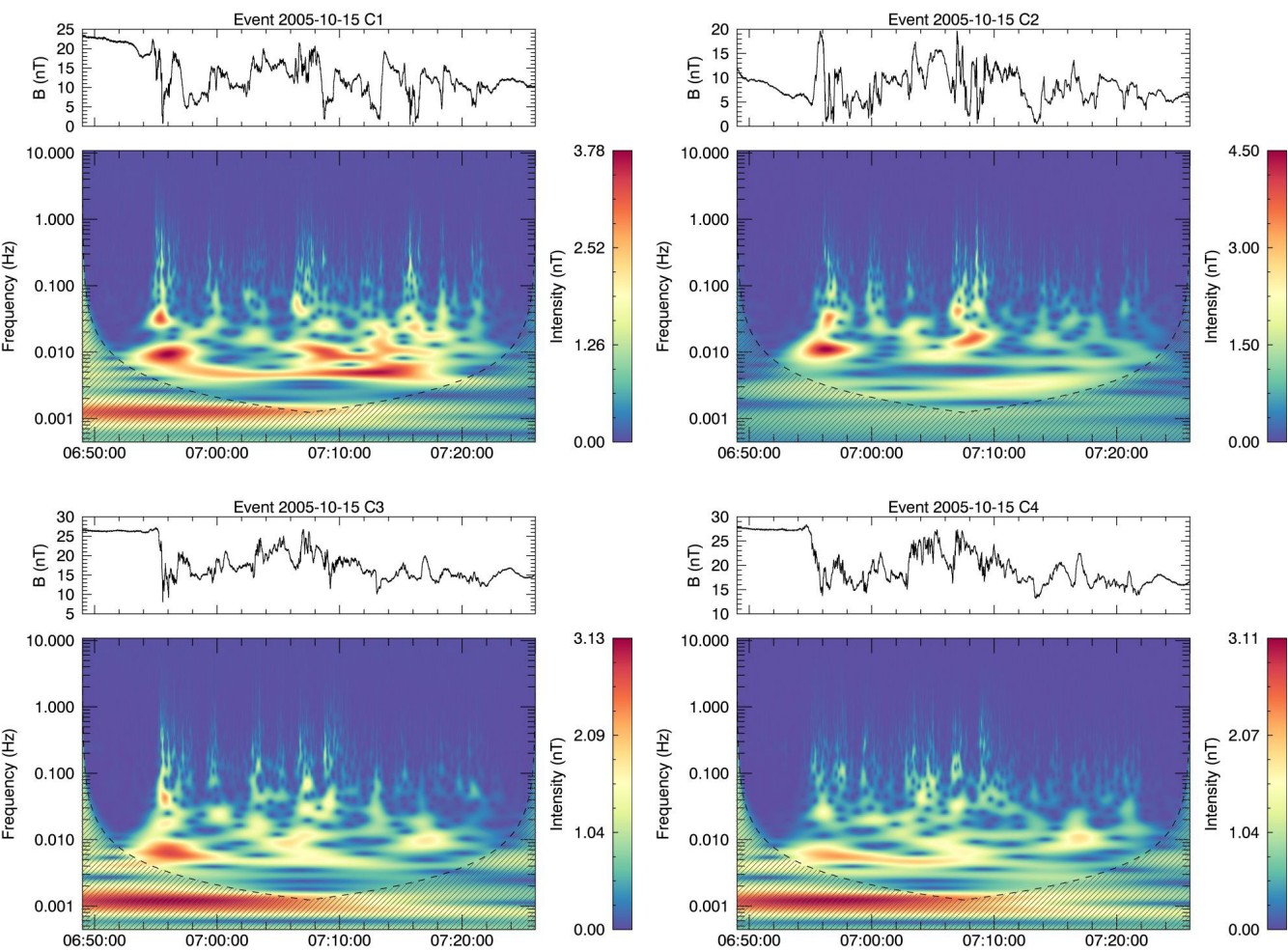

**Figure 5.** The results of wavelet analysis for event 2005-10-15. The cone of influence is shown by shaded region

the fact of coexistence of the inverse and direct cascades simultaneously starting at 13:47:30, and wherein the first one lasts for 2 minutes with frequency decrease from 0.015 Hz to 0.008 Hz and the more intense second one lasts for 2.5 minutes with slight frequency increase from 0.015 Hz to 0.02 Hz.

Fig. 5 presents the wavelet transform for event 2005-10-15. Taking into account the cone of influence (COI), there are no strong enhancements presented for C3 and C4. For both these satellites, only high-frequency short signals are present. Transformations for C1 and C2 have much richer frequency content. The component 0.008-0.01 Hz is present on all wavelet analysis, with the maximum amplitude shown at the C2 satellite. For C1 this signal has a broader structure, starting from 06:56 until 07:18, with frequency spanning from 0.005 to 0.02 Hz. Both of the satellites hold the same structure of short-term high-frequency signals up to 1 Hz.

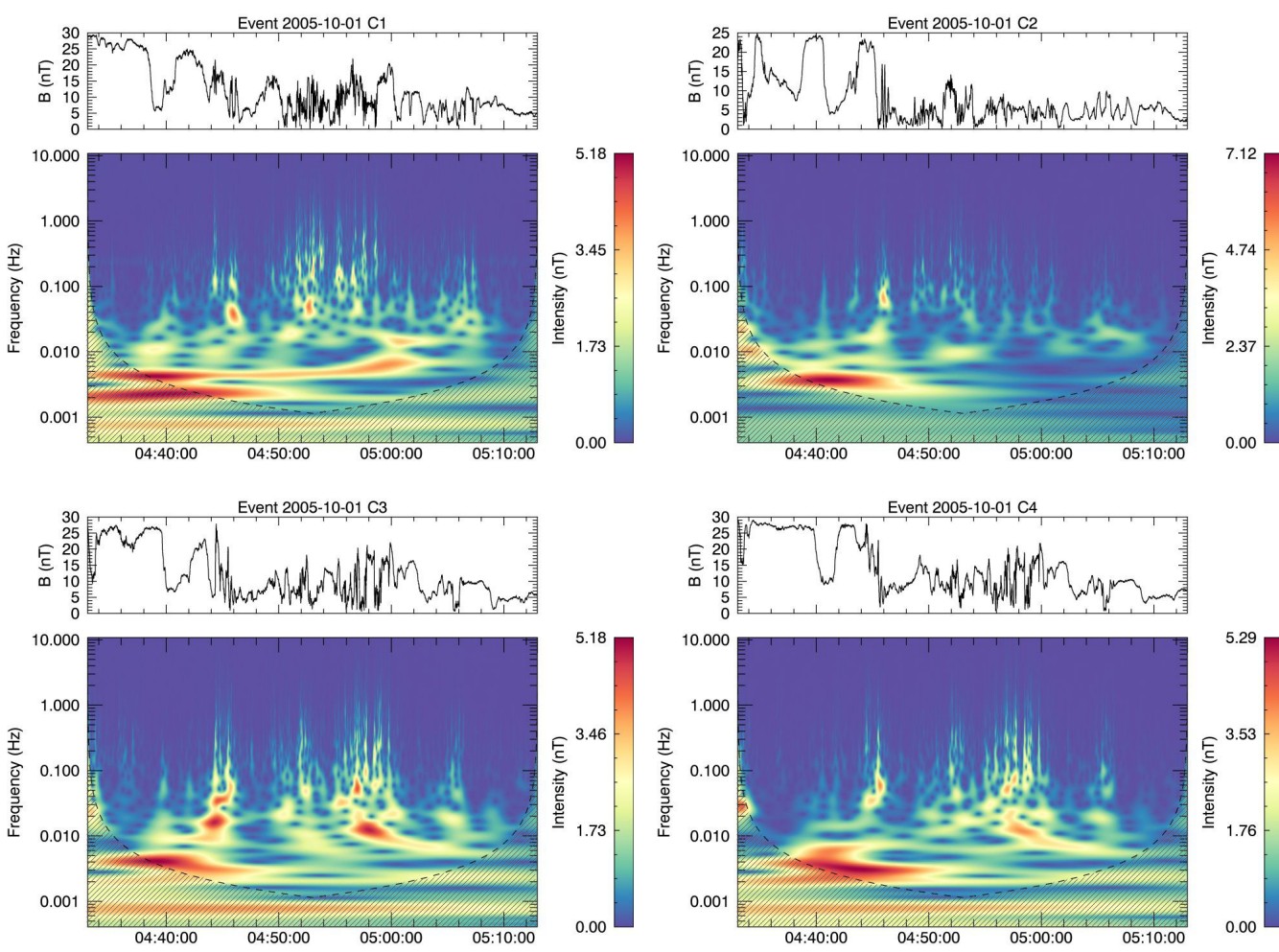

**Figure 6.** The results of wavelet analysis for event 2005-10-01. The cone of influence is shown by shaded region

Fig. 6 demonstrates the wavelet analysis for event on 2005-10-01. The 3rd and 4th SCs were located relatively close, with 1st and 2nd slightly behind in the magnetotail. Although the onset of dipolarization begins at 04:49, where $B_z$ component becomes comparable with magnetic field magnitude $B$, the signal up to this moment is not devoid of high-amplitude changes. Transforms for C3 and C4, show strong signals in the frequencies ranging from 0.002 Hz to 0.004 Hz, which span for 10 minutes, with different times for intensity maxima: 04:39 for 3rd SC and 04:42 for 4th. In both cases, a structure of inverse cascade can be traced before dipolarization onset: the frequency decreases from 0.005 to 0.002 Hz. The second inverse cascade lasts for 10 minutes starting from 04:57 in time domain with a gradual decrease in the frequency range from 0.015 to 0.005 Hz, while at higher frequency it breaks up into smaller wave forms. Wavelet decomposition for C2 differs from the others, primarily by the absence of any cascade during the turbulent phase of dipolarization with distinct component at 0.0035 Hz which spans

for 8 minutes. It is interesting to note that just for this SC the spectral slope of the PSD spectrum is less in absolute value in comparison with other SCs: $-2.5$ against $-2.8$. The relatively short components, with durations of less than 2 minutes, extend in the frequency range from 0.02 Hz up to 0.2 Hz. For C1 there is an enhancement with long duration at 0.004 Hz, which spans more than 20 minutes, and one with a gradual increase in frequency up to 0.007 Hz, i. e. direct cascade. Such prolonged intensity enhancements are observed with a wide frequency coverage beginning with 0.002 up to 0.01 Hz. A large number of high frequency components appears in measurement signals from both satellites.

Thus, during the dipolarization, the magnetometers of all spacecraft recorded powerful signals with periods of 50, 100, 125, 166, 200 sec, corresponding to Pc4 (45-150 sec) and Pc5 (150-600 sec) pulsations, as well as direct and inverse cascade processes. The presence of inverse cascade processes indicates that together with the decay of the vortex structures, self-organization also takes place, i.e. smaller vortices are grouped into larger vortices. In the analyzed events Pc pulsations observed by all satellites — in the spatial range of 11–17 $R_E$. The largest number of cascade processes is observed at a distance of 15-16 $R_E$, and the largest number of inverse cascades is in the range 13-14 $R_E$.

## 3.3   Statistical analysis

In the presence of intermittency in magnetic field fluctuations, the energy cascade is characterized by non-homogeneous non-linear transfer of energy among smaller and smaller structures, with the result of concentrating the energy on limited regions of space.

This effect becomes more and more intense at smaller and smaller scales. More properly, intermittency corresponds to scale dependent, non-Gaussian, heavy tailed probability distribution functions (PDFs) of the field fluctuations (Frisch, 1995). Non-Gaussianity of the PDFs, which increases as the spatial scale decreases, is indeed due to the presence of the intense, phase correlated fluctuations, due to the transfer of energy between contiguous eddies. It should be pointed out that spectral properties of the field are not essentially affected by intermittency. This is normally studied through the scaling properties of PDFs, or through their high order moments (the structure functions), for which models and theoretical results exist (Frisch, 1995). The observation of intermittency implies that a nonlinear, non-homogeneous energy transfer takes place in the system (Zimbardo et al., 2010).

In order to determine the presence of intermittence, an analysis on the value of the excess for all the SCs of the considered events has been performed, as well as the Hölder parameter h for spacecraft C1 has been determined. In this case, the statistical properties of the magnitude value of the magnetic field fluctuations at different time scales were analyzed. The use of the Taylor hypothesis for various regions of the magnetospheric tail is detailed in Borovsky, 2003.

The value of the kurtosis was determined by the moments of the second and fourth order from the formula by Zacks (1971):

$$K(\tau) = \frac{S_4(\tau)}{(S_2(\tau))^2} \tag{5}$$

where $S_q(\tau) = \langle |B(t+\tau) - B(\tau)|^q \rangle$ — structure function of $q$-th order, $\langle \ldots \rangle$ — time average of the data, $\tau$ — time scale (time shift), multiple of measurements discretization 0.0445 seconds. When determining the excess value of the magnetic field

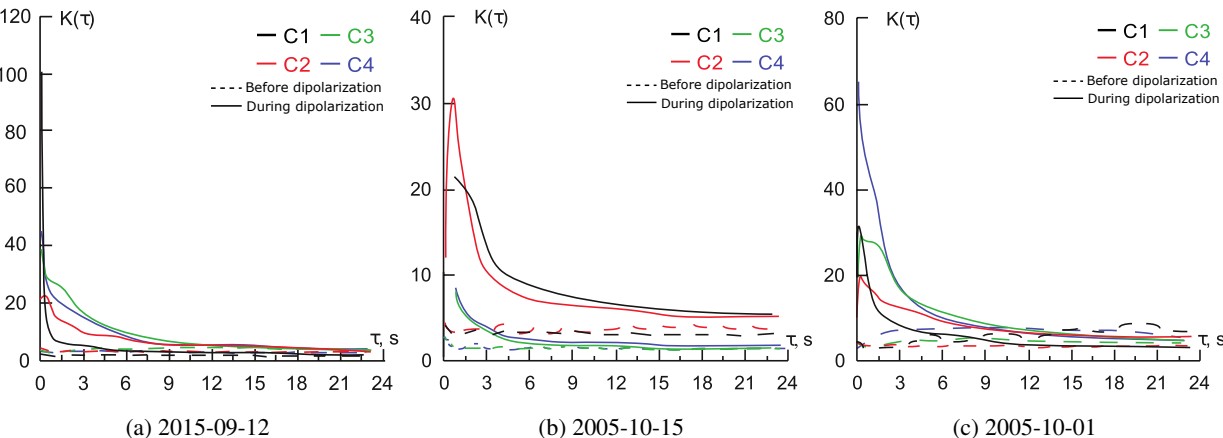

**Figure 7.** The results of kurtosis

fluctuations, the dependence of the functions $K(\tau)$ from the scale parameter $\tau$ were constructed. The significance of excesses for different mission SC and different events is shown in Fig. 7. It is clearly seen from the graphs that for the interval 1 (dotted line) for almost all satellites the value of $K(\tau)$ varies about 3 (in the range from 2 to 5), which is close to the normal distribution. The only exception is the measurement on the C1 spacecraft for 2005-10-01. Also, for interval 1, spin tone of SC

rotation is clearly observed, by sheer accident near the gyrofrequency. For the dipolarization region (interval 2, solid line), the function $K(\tau)$ on small scales varies from 100 (C1, 2015-09-12) to 8 (C3, C4, 2005-10-15).

For SC C3 and C4, changes in the value of kurtosis are very similar. The largest jump is observed for C1, 2015-09-12. A sharp drop in the kurtosis is observed on the scales to the ion-cyclotron frequency (Table 2).

The "gap" of values for interval 2 for very small $\tau$ can be explained by the instrumental error of observations.

Thus, for a region of dipolarization at small time scales, we have a distribution with a sharper vertex and broad wings (the excess value is greater than 3) than for a normal distribution.

The presence of intermittency is indicated by the analysis of the first-order structure function (Fig. 8). For a self-affine signal, $S(\tau) \approx \tau^{-h}$, where $h$ is the Hölder exponent (note, Hölder exponent is the Hurst exponent of 1st order, $h = 0.5$ for Brownian motion). The higher value of $h$ afterward indicates a persistent signal with a longer correlation than a random noise and may

imply the occurrence of reorganization during dipolarization (Chang, 1992; Consolini and Lui, 2000). In our case, the Hölder exponent is in the range $h \approx 0.659 \pm 0.005$ at the time of dipolarization.

Also, for the interval prior to dipolarization, the variations "caused" by the presence of spacecraft spin effects in the data are clearly visible.

To compare the type of turbulent processes with the available models of turbulent processes, an analysis of the high-order

structural function was done, allowing one to characterize the properties of heterogeneity at small time scales.

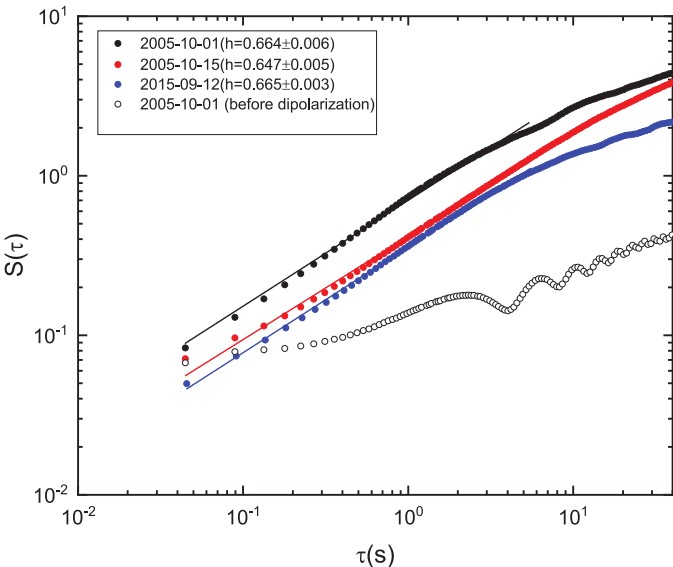

**Figure 8.** The example of Hölder exponents

In this case, the structural function is determined by the ratio:

$$S_q(\tau) = \langle |B(t+\tau) - B(t)|^q \rangle \sim \tau^{\zeta(q)} \tag{6}$$

where $\langle \ldots \rangle$ — time average of the data, $\tau$ — time shift.

The existence of the criterion of generalized self-similarity for an arbitrary pair of structural functions $S_q(\tau) \sim S_p(\tau)^{\frac{\zeta(q)}{\zeta(p)}}$ allows one to find $\zeta(q)$ and estimate the type of turbulent and diffusion processes (Dubrulle, 1994). In this case, the nonlinear functional dependence $\zeta(q)$ from the order of the moment $q$ for experimental data is a consequence of the intermittency of processes. For the interpretation of the nonlinear spectrum $\zeta(q)$, the log-Poisson model of turbulence is used, in which the power index of the structural function is determined by the relation (Dubrulle, 1994; She and Leveque, 1994; Kozak et al., 2011):

$$\zeta(q) = (1 - \Delta)\frac{q}{3} + \frac{\Delta}{1 - \beta}\left[1 - \beta^{\frac{q}{3}}\right] \tag{7}$$

where $\beta$ and $\Delta$ — parameters that characterize intermittency and singularity of dissipative processes, respectively. It is important to note that within the framework of this model a stochastic multiplicative cascade is considered, and the logarithm of dissipation energy is described by the Poisson distribution. For isotropic three-dimensional turbulence — $\Delta = \beta = 2/3$ (SL) (She and Leveque, 1994).

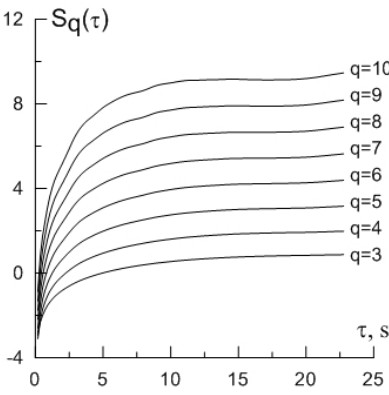

**Figure 9.** Dependence of order of structure function for different time scales during dipolarization (event 2005-10-15)

The power law of the type $S_q(\tau) \sim \tau^{\zeta(q)}$ (i.e. self-similarity - linear dependence) is observed on limited time scale intervals (Fig. 9). For the considered satellite measurements, this interval is close to the value of the ion cyclotron frequency during dipolarization (Table 2).

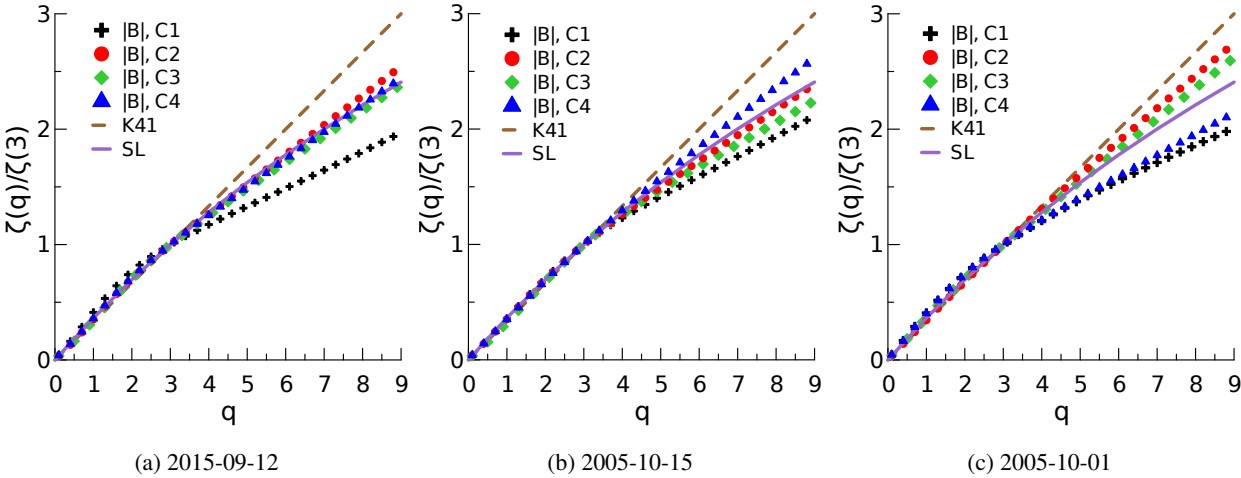

(a) 2015-09-12         (b) 2005-10-15         (c) 2005-10-01

**Figure 10.** The results of ESS-analysis (during DP). Ratio of the power of the q-th order structural function to the third order function power. The experimental data for the magnetic field are marked with symbol; the solid line corresponds to the value calculated using the formula in the log-Poisson cascade model for $\Delta = \beta = 2/3$ (SL), and the dotted line corresponds to the $q = 3$ (K41)

The results of scaling the moments of the probability density function for different orders of $q$ in the analysis of small-scale turbulence and comparing them with the Kolmogorov model are shown in Fig. 10. The results of the ESS analysis of the satellite measurements indicate the heterogeneity of turbulent processes during the dipolarization to describe what can be a log-Poisson cascade model with fitting parameters. The obtained values of the parameters $\beta$ and $\Delta$ are given in Table 4. In addition, the obtained values can be used to determine the characteristics of the diffusion transfer of plasma. In this case, the properties

**Table 4.** ESS-analysis parameters and diffusion coefficients

| Event | SC | $\beta$ | $\Delta$ | $R(-1) = \Delta[1/\beta - 1]$ |
|-------|-----|---------|----------|-------------------------------|
|  | C1 | $0.5 \pm 0.029$ | $0.77 \pm 0.026$ | 0.77 |
| 2015-09-12 | C2 | $0.6 \pm 0.018$ | $0.45 \pm 0.019$ | 0.30 |
|  | C3 | $0.52 \pm 0.091$ | $0.43 \pm 0.009$ | 0.40 |
|  | C4 | $0.58 \pm 0.016$ | $0.46 \pm 0.014$ | 0.33 |
|  | C1 | $0.51 \pm 0.015$ | $0.67 \pm 0.012$ | 0.64 |
| 2005-10-15 | C2 | $0.68 \pm 0.025$ | $0.72 \pm 0.019$ | 0.34 |
|  | C3 | $0.34 \pm 0.027$ | $0.22 \pm 0.026$ | 0.43 |
|  | C4 | $0.51 \pm 0.026$ | $0.24 \pm 0.028$ | 0.23 |
|  | C1 | $0.45 \pm 0.015$ | $0.41 \pm 0.013$ | 0.5 |
| 2005-10-01 | C2 | $0.51 \pm 0.013$ | $0.2 \pm 0.021$ | 0.2 |
|  | C3 | $0.45 \pm 0.024$ | $0.21 \pm 0.019$ | 0.26 |
|  | C4 | $0.51 \pm 0.026$ | $0.54 \pm 0.018$ | 0.52 |

of diffusion are considered within the concept of a multi-fractal multiplicative cascade (Lovejoy et al., 1998). The coefficient of generalized diffusion is determined by the parameters of the structural function $\zeta(q)$ (intermittency and singularity) by the relations by Lovejoy et al. (1998); Prokhorenkov et al. (2015):

$$D \propto \tau^R, \qquad R = \tilde{R}(-1), \qquad \tilde{R}(q) = q - \zeta(3q), \qquad R = \Delta(1/\beta - 1) \tag{8}$$

5      This approach is used to estimate the transfer in a statistically inhomogeneous medium, and the index $R$, in general, is determined by the fractal properties of the medium and characterize (on average) the topological properties (connection properties that determine the transfer) of a stochastic structure of turbulence.

     The resulting values of $R$ lie within the range from 0.20 to 0.77 (Table 4). Given that the law of particle displacement over time is given by the formula by Treumann et al. (1990); Chechkin et al. (2008); Zaburdaev et al. (2015): $\langle \delta x^2 \rangle \propto D\tau \propto \tau^\delta$

10    with an indicator $\delta \propto 1 + R \approx 1.20$–$1.77 > 1$, this dependence means the existence of super-diffusion.

## 4   Conclusions

As a result of the analysis, it can be concluded that the relative variations of the magnetic field during the dipolarization exceed the value before dipolarization by more than 5 times. The distribution functions of magnetic field fluctuations during the

disruption of the current layer indicate the non-Gaussian statistics of processes, as well as the excess of large-scale perturbations generated by the source.

Comparing the structure functions of the magnetic field fluctuations during dipolarization with Kolmogorov model, it is impossible to describe turbulent processes on small time scales using homogeneous model. Using the coefficients of intermittency and singularity of turbulent processes found in the ESS analysis, the power law of the generalized diffusion coefficient on the scale was obtained (the power index varies within the range from 0.2 to 0.77), indicating the presence of super-diffusion processes.

One of the important results is the significant difference of the spectral indices for the intervals before and during the dipolarization. Before dipolarization the spectral index lies in the range from $-1.68 \pm 0.05$ to $-2.08 \pm 0.05$ ($\sim 5/3$ according to the Kolmogorov model), and during dipolarization the type of turbulent motions changes: on large time-scales the turbulent flow is close to the to the homogeneous models of Kolmogorov (1941) and Iroshnikov-Kraichnan (1959) (the spectral index lies in the range from $-2.20$ to $-1.53$), and at smaller time scales the spectral index lies in the range from $-2.89$ to $-2.35$ (the Hall-MHD model). The kink frequency is less or close to the average value of the proton gyrofrequency. The Hall-MHD model includes the Hall term in the magnetic induction equation. The Hall term is proportional to the ion inertial length $c/\omega_{pi}$, which means this term is important for the small scales (Galtier and Buchlin, 2007). Both the standard MHD and the electron MHD can be recovered from Hall-MHD by taking appropriate limits. By considering magnetic turbulence spectra for scales smaller than $c/\omega_{pi}$, Galtier and Buchlin (2007) found a number of spectral indexes, which go from $\alpha = 7/3$ when magnetic energy dominates kinetic energy, to $\alpha = 11/3$ when kinetic energy dominates magnetic energy.

Also, within the framework of the research the following results were obtained:

– the higher the PSD value, the greater is the value of the height of the excess;

– log-Poisson mode of turbulent processes with She-Leveque parameters corresponds to variations in the value of $K(\tau)$ in the range of $30 - 40$;

– the spectral indices correlate with the values of the diffusion coefficient.

The wavelet analysis showed the presence of both direct and inverse cascade processes, as well as the presence of Pc pulsations. The presence of Pc pulsations in the region of dipolarization was also discussed in Panov et al. (2013, 2015).

Thus, during dipolarization the large-scale and multi-fractal disturbances of the magnetic field are observed and the presence of inverse cascade processes also indicates the possibility of self-organization processes.

*Acknowledgements.* The work was conducted in the frame of complex program of National Academy of Science of Ukraine in Plasma Physics; with support of education program of Ministry of Education and Science of Ukraine No 2201250 "Education, Training of students, PhD students, scientific and pedagogical staff abroad"; the grant Az. 90 312 from the Volkswagen Foundation ("VW- Stiftung") and International Institution of Space Research (ISSI-BJ).

We acknowledge Cluster Science Archive (https://www.cosmos.esa.int/web/csa), PI and teams of FGM and CIS instruments for providing the data. The Cluster data used in this study were downloaded from the Cluster Science Archive version 1.2.1 at https://www.cosmos.esa.int/web/csa.

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
