# Peer review of "Turbulent Processes in the Earth's Magnetotail: Spectral and Statistical Research"

_Annales Geophysicae, 2018_

## Referee Comment (RC1) · G. Zimbardo (Referee) · 9 Jul 2018

Report on "Turbulent Processes in the Earth's Magnetotail: Spectral and Statistical Research" by Kozak et al.

This manuscript reports a detailed analysis of magnetic fluctuations in the Earth's magnetotail, on occasion of Cluster spacecraft observations of three dipolarization events. Several techniques are used to assess the turbulence properties, like the computation of the power spectral density, the wavelet spectra, the structure functions and the computation of the kurtosis. It is found that turbulence is much stronger after the dipolarization events, and that both a direct and an inverse turbulent cascade are possible. The manuscript is interesting and potentially appropriate for publication on Annales
Geophysicae, but the following points whould be taken into account:

1. In Figure 1, only the magnitude of the magnetic field is given, however, to better appreciate what is seen by spacecraft, it would be better to give also the components of B in the GSM system, at least for the Cluster spacecraft closer to the current sheet center.

2. On page 7, line 20, the authors refer to the effect of heavy ions on the power spectral break. Can they show the presence of heavy ions in the plasma data, e.g., from the CIS instrument?

3. In the middle left panel of Figure 3, the PSD measured by C2 is more than 1 order of magnitude larger than that measured at the other spacecraft. Why? This is not seen after dipolarization: why?

4. Page 15: is the Holder exponent h the same as the Hurst exponent?

5. I presume that, in Eq. (6),  $B(\tau)$  should be B(t).

6. The discussion of superdiffusion on pages 16-17 is not very clear: superdiffusion of what, plasma? energetic particles? fluid particles? The most reliable model of superdiffusion is based on Levy walks, see the recent review by Zaburdaev et al., Rev. Mod. Phys., 2015.

7. For many calls to references, the names appear out of the parentheses when they would be expected to be inside the parentheses. In LaTeX, this amounts to changing from citet to citep. On the other hand, some reference calls are correct as they are.

8. The English is good but not perfect. Proof reading by someone more fluent in English is suggested.

ANGEOD

---

## Referee Comment (RC2) · Anonymous Referee #2 · 13 Jul 2018

The manuscript addresses the turbulence processes in the magnetotail using Cluster-2 observations. Spectral, wavelet, and statistical analyses are conducted particularly for the dipolarization events. Significant difference of the spectral indices is found for the intervals before and during the dipolarization. Before dipolarization the spectral index $\sim 5/3$ according to the Kolmogorov model. During the depolarization the type of turbulence motion changes, and a spectral break is present around the proton gyrofrequency. The presence of both direct and inverse cascade process is explained as the possibility of self-organization processes. I find the paper is very interesting. The results are sufficiently new. But I have the following comments and questions that need to be considered before the manuscript can be published.

1. Line 26: "Although magnetic reconnection is not a plasma instability process, it

requires an instability such as ion tearing instability". This statement is not true.

2. In the Introduction, it is described that the processes of current disruption (CD) and magnetic reconnection are very different in their structure and dynamics. In general, plasma turbulence is important not only to the current disruption but also present in reconnection. For the 3 selected dipolarization events, the paper attributes the event of 2015 to CD. What about the other 2 events? Can you characterize the differences in the turbulence spectra/signatures between the dipolarization structures in CD and reconnection?

3. Page 3, last paragraph: what described in this paragraph for Figure 1 are not shown in Figure 1 (e.g., no components of B are plotted).

4. Dipolarization is a kinetic structure, and the existence of spectral break around the ion cyclotron frequency shows evidence of wave-particle interaction. But the study does not address the turbulence spectrum and direct/inverse cascade from the kinetic point of view. Can you provide analysis for the spectra parallel and perpendicular to the magnetic field?

5. The way the citations are quoted needs to be fixed. They are not at the right places or in the right format. Some other symbols are also messed up. For example, in the description of the range of exponents.

---

## Short Comment (SC1) · 19 Jul 2018

Comments on "Turbulent Processes in the Earth's Magnetotail: Spectral and Statistical Research" by Kozak et al. The authors used Cluster data to investigate turbulent properteis before and during the dipolarization process in the near Earth's magnetotail. This work is interesting, but more work is needed to do. The comments below are helpful to improve this paper.

Q1. The authors used most of parts to introduce the sub-storm in the magnetotail, but just simply mentioned several works which are related with plasma turbulence in the magnetotail. However, there are a lot of important works which have investigated the turbulent process (such as spectral index, intermittence, multifractal etc.) associated

[Figure]

with BBF, magnetic reconnection, or dipoalrization in the magnetotail plasma sheet. I strongly suggest the authors to introduce the recent progress on this topic and compare with their results in this paper. Some related references are shown below: Vörös, Z., et al. (2003), Multiâ̆scale magnetic field intermittence in the plasma sheet, Ann. Geophys., 21(9), 1955–1964. Vörös, Z., et al. (2004), Magnetic turbulence in the plasma sheet, J. Geo- phys. Res., 109, A11215, doi:10.1029/2004JA010404. Vörös, Z., W. Baumjohann, R. Nakamura, A. Runov, M. Volwerk, H. K. Schwarzl, A. Balogh, and H. Rème (2005), Dissipation scales in the Earth's plasma sheet estimated from Cluster measurements, Nonlinear Process. Geophys., 12(5), 725–732. Vörös, Z., W. Baumjohann, R. Nakamura, M. Volwerk, and A. Runov (2006), Bursty bulk flow driven turbulence in the Earth's plasma sheet, Space Sci. Rev., 122(1–4), 301–311, doi:10.1007/s11214-006-6987-7. Vörös, Z., W. Baumjohann, R. Nakamura, A. Runov, M. Volwerk, T. Takada, E. A. Lucek, and H. Reme (2007), Spatial structure of plasma flow associated turbulence in the Earth's plasma sheet, Ann. Geophys., 25 (1), 13–17. Huang, S. Y., M. Zhou, X. H. Deng, et al. (2012), Kinetic structure and wave properties associated with sharp dipolarization front observed by Cluster, Ann. Geophys., 30, 97-107. Klimas, A., V. Uritsky, and E. Donovan (2010), Multiscale auroral emission statistics as evidence of turbulent reconnection in Earth's midtail plasma sheet, J. Geophys. Res., 115, A06202, doi:10.1029/2009JA014995. Huang, S. Y., M. Zhou, F. Sahraoui, et al. (2010), Wave properties in the magnetic reconnection diffusion region with high $\beta$: Application of the k-filtering method to Cluster multispacecraft data, J. Geophys. Res., 115, A12211, doi:10.1029/2010JA015335. Huang, S. Y., M. Zhou, F. Sahraoui, A. Vaivads, X. H. Deng, M. André, J. He, H. Fu, H. M. Li, Z. Yuan, and D. D. Wang (2012), Observations of turbulence within reconnection jet in the presence of guide field, Geophys. Res. Lett., 39, L11104 doi:10.1029/ 2012GL052210. Weygand, J. M., et al. (2005), Plasma sheet turbulence observed by Cluster II, J. Geophys. Res., 110, A01205, doi:10.1029/2004JA010581. Weygand, J. M., M. G. Kivelson, K. K. Khurana, H. K. Schwarzl, R. J. Walker, A. Balogh, L. M. Kistler, and M. L. Goldstein (2006), Nonâ̆ selfâ̆similar scaling of plasma sheet and solar wind

probability distribu- tion functions of magnetic field fluctuations, J. Geophys. Res., 111, A11209, doi:10.1029/2006JA011820.

Q2.  Line #23 in Page 3, 3 events cannot cover from 2005 to 2015.  After read the paper, one knows two events in 2005, one event in 2015. Thus, I suggest the authors to re-write this sentence to avoid the confusedness.

Q3.  Line # 30-32 in Page 3 and Line #1-6 in Page 4, the authors describe three components and depolarization, but they didn't show any components in Figure 1. As a reader, I strongly suggest the authors present the components of magnetic field.

Q4.  Table 1 in Page 5.  The authors show the features of the depolarization front. The speed of the DF and thickness are estimated by timing analysis.  However, the separation of four Cluster spacecraft is about 2 Re in 2005. This separation is able to compare with the dawn-dusk scale of DF (Fu et al., 2012; Huang et al., 2015), which leads to that the timing results may be not correct. One should be careful to perform timing analysis such situation.  Fu, H. S., Y. V. Khotyaintsev, A. Vaivads, M. André, and S.Y. Huang (2012), Electric structure of dipolarization front at sub-proton scale, Geophys. Res. Lett., 39, L06105, doi:10.1029/ 2012GL051274. Huang, S. Y., et al. (2015) Dawn-dusk scale of dipolarization front in the Earth's magnetotail: multi-cases study, Astrophys Space Sci, 357, 22, doi:10.1007/s10509-015-2298-3

Q5. Table 3 in Page 8: Kink frequency? I can't see obvious kinks in the spectrum in Figure 3. Thus, I would like to use the frequency of breakpoint to replace it.

Q6. Line #12 in Page 10: Inverse and direct cascade. What's inverse and direct cas- cade? What's the definition of two cascades? How to identify two different cascades? I think such introduction will make the paper more clearlyz.

Q7. Line #8 in Page 13: It should be pointed out that spectral properties of the field are not sensitively affected by intermittency. Any references to support this?

Q9.  Line #7-8 in Page 3: "This allows one to get an idea of the physical properties of plasma turbulence and a description of the transport processes in the turbulent regions in qualitative and quantitative term" There are also a lot of references to qualitatively and quantitatively investigate the turbulent in the magnetosheath or solar wind. Breuillard, H., Yordanova, E., Vaivads, A., et al. 2016, The Effects of Kinetic Instabilities on Small-scale Turbulence in Earth's Magnetosheath, ApJ, 829, 54 Huang, S. Y., F. Sahraoui, X. H. Deng, et al. (2014), kinetic turbulence in the terrestrial magnetosheath: cluster observations, Astrophys. J. Lett., 789, L28 Sahraoui, F. S., Huang, Y., De Patoul, J., et al. 2013, Scaling of the electron dissipation range of solar wind turbulence, ApJ, 777, 15
 He, J. S., Tu, C., Marsch, E., & Yao, S. 2012, Do oblique alfvén/ion-cyclotron or fast-mode/whistler waves dominate the dissipation of solar wind turbulence near the proton inertial length? ApJL, 745, L8
 Sahraoui, F., Belmont, G., Rezeau, L., et al. 2006, Anisotropic Turbulent Spectra in the Terrestrial Magnetosheath as Seen by the Cluster Spacecraft, PhRvL, 96, 075002 Huang, S. Y., L. Z. Hadid, F. Sahraoui, Z. G. Yuan, and X. H. Deng (2017), On the Existence of the Kolmogorov Inertial Range in the Terrestrial Magnetosheath Turbulence, Astrophys. J. Lett., 836, L10, doi.org/10.3847/2041-8213/836/1/L10

Q10. I suggest the authors compare their results with previous observations, and discuss them.

Q11. Some typos in the Line #12-13 in Page 1 and line #21-22 in Page 7: $-2.20$ Ãů $-1.53$, $-2.89$ Ãů $-2.35$ in Line #1-2 in Page 18: 0.20 Ãů 0.77 1.20 Ãů 1.77 in Line #17-18 in Page 18: -2.20 Ãů 1.53 2.89 Ãů 2.35

---

## Short Comment (SC2) · 20 Jul 2018

Dear referee,

In the addition to my previouws answer I want to add the more detail information about the paper by Ted Speiser about reconnection without the need for (noise) turbulence.

T. W. Speiser Conductivity without collisions or noise // Planet. Space Sci. 1970, Vol. 18, pp. 613 to 622.

Best regards, Liudmyla

---

## Author Comment (AC1) · 20 Jul 2018

Dear Kui Jiang,

We appreciate your interest in our publication

We give short answers to the topics raised in the discussion of the publication:

1. The work is a research, and not an overview and we think, that the authors think that the information in the article is sufficient and the readers of the article are sufficiently erudite to evaluate the results obtained in the framework of the conducted researches with the studies carried out by other authors and in various fields of the magnetosphere. Moreover, the list proposed in the comments can be expanded.

2. It is noted in the article that this is three events! Although we do not mind writing three articles analyzed in the text of the article (two events of 2005 and one event in 2015).

3. Figure of the components of the magnetic field for the satellites, closest to the current layer in GSM, added in Fig. 1 (Fig. 1b)

4. Estimates of the parameters of the dipolarization front were carried out precisely on the basis of the material of the article Fu, HS, Khotyaintsev, YV, Vaivads, A., André, M., and Huang, SY: Occurrence rate of earthward propagating dipolarization fronts, Geophysical Research Letters, 39, https://doi.org/10.1029/2012gl051784, 2012 of which you are mentioning. The link to the article is also in Table 1 and in the text and in the list of references.

5. The kink frequency and frequency of breakpoint are synonyms.

6. In the article there is a feature of direct and inverse cascades, for example on page 10, line 31: "In both cases, a structure of inverse cascade can be traced before dipolarization onset: the frequency decreases from 0.005 to 0.002 Hz."

7. In this article, in that paragraph there are reference to "Frisch, U.: Turbulence. The legacy of A. N. Kolmogorov., Cambridge University Press, 1995." and "Zimbardo, G., Greco, A., Sorriso-Valvo, L., Perri, S., Vörös, Z., Aburjania, G., Chargazia, K., and Alexandrova, O.: Magnetic Turbulence in the Geospace Environment, Space Science Reviews, 156, 89–134, https://doi.org/10.1007/s11214-010-9692-5, 2010.".

8. In the paper, the analysis of diffusion processes is carried out particularly at the Earth's magnetotail (magnetosheet). Therefore, we do not have comparisons with the diffusion processes in the other part of the magnetosphere - the magnetosheath, as well as in the plasma of the solar wind. References that you specify refers to other regions.

9. The purpose of the work was not to compare the diffusion coefficients in different

regions of the magnetosphere and the plasma of the solar wind.

10. We didn't find listed typos in the presented PDF to the Journal.

Best regards, Authors.
* * *
[Figure]

**Fig. 1.** Figure 1b. Example of fluctuations of magnetic field.

---

## Author Comment (AC2) · 20 Jul 2018

The authors are grateful to the anonymous reviewer for his/her comments and attention to this publication.

1. Magnetic reconnection is not an instability because it does not have a growth rate and an associated frequency with it. These two parameters are the characteristics of an instability as illustrated in any textbook that describes instability. If the referee thinks that our statement is not true, then how does the referee define an instability?

2. Magnetic reconnection (MR) can involve turbulence, but it is not essential. Ted Spicer had a paper describing MR without noise (i.e., turbulence) based on chaotic particle motions around the X-line to break the frozen-in condition. Since we are

addressing CD and not MR, we shall defer the discussion on the turbulence spectra/signature in MR and their differences with CD in a future study when MMS team reports MR events in the tail. The event on 2015 satisfies the most the conditions for the CD model. (Just in case, we have added "most" to the article in the sentence on page 3, line 27 (old version)).

3. Figure of the components of the magnetic field for the satellites, closest to the current layer in GSM, added in Fig. 1 (Fig. 1b). The description on p.3 describes the characteristics of CD and figure 1 shows the magnetic disturbances for the 3 events under study. The criteria for CD in these 3 events are checked out.

4. The kinetic nature of dipolarization is well demonstrated by the activities in scales not describable by any fluid approach as well as by the evolution of small scale features. These activities provide clues to the underlying kinetic process. This study does not include kinetic analysis of what the CD process is.

5. We have reviewed and improved the language and typesetting of the article.

Best regards, Authors.

Please also note the supplement to this comment:
https://www.ann-geophys-discuss.net/angeo-2018-50/angeo-2018-50-AC2-supplement.pdf

**Supplement:**

**Turbulent Processes in the Earth's Magnetotail: Spectral and Statistical Research**

Liudmyla Kozak1,2, Bohdan Petrenko1, Anthony T.Y. Lui3, Elena Kronberg4,5, Elena Grigorenko6, and Andrew Prokhorenkov1

1Taras Shevchenko National University of Kyiv, Kyiv, Ukraine

[revised manuscript text omitted]

---

## Author Comment (AC3) · 20 Jul 2018

Dear Professor Zimbardo,

We are grateful for the comments and advices on the article.

We took into account your remarks and clarifications.

1. Fig. 1 is supplemented by the components of the magnetic field in the GSM coordinate system for one of the spacecraft (the SC that is closest to the current layer) for each of the events considered (Fig. 1b).

2. According to the available measurements of CIS instrument we estimate the percentage content of heavy elements in the region of the magnetic field dipolarization.

The text in the brackets has been added.

"according to the measurements of the CIS instrument for the event 2005-10-01, in the region of the magnetic field dipolarization, the percentage of oxygen ions in relation to protons ($\langle n(O^+)\rangle/\langle n(H^+)\rangle$) is $21.1 \pm 10.0\%$ (SC C3) and $9.3 \pm 1.5\%$ (SC C4), and the percentage of helium in relation to protons ($\langle n(He^+)\rangle/\langle n(H^+)\rangle$) $\sim 2.4 \pm 0.3\%$ (SC C3) and $\sim 4.8 \pm 0.7\%$ (SC C4); for the event 2005-10-15 — $\langle n(O^+)\rangle/\langle n(H^+)\rangle \sim 11.1 \pm 1.0\%$ (SC C4), and $\langle n(He^+)\rangle/\langle n(H^+)\rangle \sim 3.4 \pm 0.5\%$ (SC C4); for the 2015-09-12 event — $\langle n(O^+)\rangle/\langle n(H^+)\rangle \sim 18.9 \pm 7.3\%$ (SC C4), and $\langle n(He^+)\rangle/\langle n(H^+)\rangle \sim 15.8 \pm 5.4\%$ (SC C4)"

3. As shown in Figure 1 in the middle panel (event for 2005-10-15), the level of fluctuations of the magnetic field module in the interval 1 (that is, in the prepolarization interval) for C2 exceeds the value of the level of fluctuations for other spacecrafts, which is manifested in higher values of the PSD. After the start of dipolarization (interval 2), the level of field fluctuations for different spacecrafts is not very different from each other, and in addition, the relative uniformity of the spectrum value is amplified by its "blurring" on a large scale.

4. Holder's exponent h is indeed the Hurst index of the 1st order: h = H(1). The text of the article has been added (Holder exponent is the Hurst exponent of 1st order)

5. Thank you, indeed, in equation 6 we replaced B(t) with B ($\tau$).

6. Thank you for the information on review by Zaburdaev et al., Rev. Mod Phys. 2015. It will be useful for our further research and we included it in our references. It should be noted that in this paper we basically took the relation between the exponent of the structural function and the generalized diffusion coefficient, which, if we are not mistaken, was first obtained in detail in Lovejoy 1998. (We refer to this article in the references) Usually there is a connection between the definition of the diffusion coefficient considered in the "Levy flights" analysis and in the framework of the ESS analysis. In our 2015 work (Kozak, L. V., Prokhorenkov, A. S. Savin, S. P. Statistical

analysis of the magnetic fluctuations in boundary layers of Earth's magnetosphere. Adv. Sp. Res. 56, 2091–2096 (2015)) for the area of the magnetosphere, we compared the values of the diffusion coefficient obtained by two different approaches.

Since in this paper we evaluate the diffusion coefficient from the analysis of the magnetic field fluctuations, we obtain a coefficient which characterizes the transport processes associated with the spatial-temporal structure of the magnetic turbulence.

We have added in the article: "In this case, the properties of diffusion are considered within the concept of a multi-fractal multiplicative cascade" The equation (8) described in more details and the sentences are supplemented with: "... fractal properties of the medium and characterize (on average) the topological properties (connection properties that determine the transfer) of a stochastic structure of turbulence"

7. Thank you. Corrections have been made.

8. The article text has been submitted to the professional translator.

Regards, Authors

Please also note the supplement to this comment:
https://www.ann-geophys-discuss.net/angeo-2018-50/angeo-2018-50-AC3-supplement.pdf

---

## Short Comment (SC3) · 22 Jul 2018

[revised manuscript text omitted]

---

## Author Response (AR1)

In the updated version of the article we took in account comments from the reviewers.
In particular,
- We added Fig. 1b, description of the concentration of heavy ions, citation on the article Lovejoy, 1998 and Zaburdaev, 2015;
- We puted more details of Hurst exponent and diffusion coefficient;
- We changed range symbol into "from … to …" and changed 2005 - 2015 into the three events (two events in 2005 and one event in 2015);
- We made the stylistic and gramatic correction, edits regarding citation to articles.

All changes are marked in color.

Also the comments of the Topical Editor were taken into account
1.    In figures Fig. 4 - Fig. 6, COI region have been shaded.
2.    We added to the article information as well as a reference to the article by Speiser, 1970.
All changes are marked in color as well.

Best regards,
Kozak Liudmyla